# Black Silicon: Breaking through the Everlasting Cost vs. Effectivity Trade-Off for SERS Substrates

**DOI:** 10.3390/ma16051948

**Published:** 2023-02-27

**Authors:** Lena Golubewa, Hamza Rehman, Yaraslau Padrez, Alexey Basharin, Sumit Sumit, Igor Timoshchenko, Renata Karpicz, Yuri Svirko, Polina Kuzhir

**Affiliations:** 1Department of Molecular Compound Physics, State Research Institute Center for Physical Sciences and Technology, Sauletekio Av. 3, 10257 Vilnius, Lithuania; 2Department of Physics and Mathematics, Center for Photonics Sciences, University of Eastern Finland, Yliopistokatu 7, 80101 Joensuu, Finland

**Keywords:** black silicon, surface-enhanced Raman spectroscopy, surface plasmon resonance, nanostructures, etching

## Abstract

Black silicon (bSi) is a highly absorptive material in the UV-vis and NIR spectral range. Photon trapping ability makes noble metal plated bSi attractive for fabrication of surface enhanced Raman spectroscopy (SERS) substrates. By using a cost-effective room temperature reactive ion etching method, we designed and fabricated the bSi surface profile, which provides the maximum Raman signal enhancement under NIR excitation when a nanometrically-thin gold layer is deposited. The proposed bSi substrates are reliable, uniform, low cost and effective for SERS-based detection of analytes, making these materials essential for medicine, forensics and environmental monitoring. Numerical simulation revealed that painting bSi with a defected gold layer resulted in an increase in the plasmonic hot spots, and a substantial increase in the absorption cross-section in the NIR range.

## 1. Introduction

Silicon is the most widely used electronics and photonics semiconducting material. Although a polished silicon possesses a high reflectivity in the ultraviolet (UV) [1], visible (vis) and near infrared (NIR) range [2], the micro-structuring of silicon may make it remarkably absorptive [3,4,5]. As a result, the micro-structured silicon surface acquires a deep black color, and it is often referred to as “black silicon” (bSi).

Having been discovered accidentally, bSi was initially considered an unwanted side-product [6]; however, nowadays it is undergoing a rebirth because of new opportunities it opens in a number of applications. The surface of bSi may comprise of various shapes and geometries, including needles, pyramids, cones, pillars, holes, columns, etc., which enhance absorption and suppress light scattering [7] via effective light trapping [8]. The suppression of specular reflectance and enhancement of absorbance can be described in terms of the structural defects in the Si lattice or impurities (specifically sulfur) which appear during surface micro-structuring [7]. These defects and impurities create IR absorbing states in the band gap and are also responsible for the trapping of photoexcited carriers (for flat silicon, the band gap is 1.07 eV) [9]. The absorbance caused by these defects is enhanced by multiple reflections between the microstructures of bSi, which, in turn, suppresses specular reflection and increases absorbance.

The significant increase in absorbance in UV-vis and NIR spectral ranges by bSi, compared to bulk silicon, spans the applications of bSi to many areas, including solar cells [10,11], photo-electrocatalysis [12,13], visible light and NIR sensors [9,14]. Moreover, there is also a trend to replace indium gallium arsenide and germanium in commercial NIR photodetectors [15] with bSi because it is cost-effective and enables integration of the photodetectors into a silicon platform. 

The high absorptive properties of bSi in a wide spectral range are also beneficial for nanoplasmonic applications, including surface enhanced Raman spectroscopy (SERS) [16], which employs the enhancement of the light field in the vicinity of the metal surface, where the eigenfrequency of the electron ensemble match the light frequency. If the molecules of the analyte are situated in the vicinity of the interface between the metal plasmonic particle and the environment, the surface plasmon resonance (SPR) results in an increase in the Raman scattering cross-section by several orders of magnitude, allowing one to detect trace amounts of molecules adsorbed onto the interface [17,18]. When using bSi as a supporting material for the formation of plasmonic nanostructures, significant UV-vis–NIR absorption and the phenomenon of light-trapping by the micro- and nano-structured surface of bSi may drastically increase the local electromagnetic field enhancement on the metal–environment interface. This, in its turn, will cause the increase in the Raman signal of the adsorbed molecules of the analyte.

The SERS approach is a reliable method for detecting low concentrations of molecules [19], investigating molecule–molecule [20] or molecule–surface [21] interactions, investigating living cells functions [22] and detecting pesticides [23] and poisons [24], which obviously extends its applications from fundamental science to practice, including biomedicine, forensics, environment, security, etc. However, obtaining very specific information from analytes in the form of a ‘fingerprint’ vibrational spectra, SERS exhibits some drawbacks, which significantly reduce its applications in real life. Namely, SERS substrate fabrication always presents a trade-off between stability, cost, efficiency, reliability, uniformity, density of hot-spots and scalability. Commonly used approaches for SERS substrate fabrication are based either on colloidal solutions of noble metal nanoparticles or nanostructured surfaces of noble metal slabs and films. The former usually give significant enhancement of Raman intensity; however, they are non-stable [25], storage condition-dependent [26], inconvenient in routine use [27], require immobilization on the flat surface for measuring [28], poorly reproducible and non-uniform [27,28]. The latter allows one to control the density of hot spots and to be fabricated in large areas, but they are expensive and give rather limited signal enhancement.

Pronounced surface micro-structuring of bSi in combination with exceptional absorbance may serve as a regular, stable and highly uniform support for plasmonic nanostructures, ensuring a strong Raman signal enhancement. However, strong dependence of SPR on the geometry of plasmonic structures [29,30,31] and their three-dimensional organization implies bSi profile control.

Nano- and micro-structured bSi can be fabricated by metal-assisted chemical etching, femtosecond laser assisted fabrication [8,32], laser chemical etching [7], wet chemical etching and reactive ion etching (RIE) [33]. The latter is the most suitable option when aiming for uniform surface structure coverage over large areas. The RIE system uses a plasma source consisting of highly reactive ion species, and when they bombard the sample, a chemical reaction takes place that selectively erodes away the sample surface [34].

In the present study, we fabricate bSi by using room temperature (RT) Inductively Coupled Plasma (ICP) RIE in such a way that the created surface profile supports a gold plasmonic layer of a thickness as low as almost ten nanometers with distributed cracks that still provides satisfactory Raman signal enhancement. Fabrication by RT ICP-RIE, instead of cryogenic ICP-RIE, and an extremely reduced defected gold layer allowed for a significant reduction in the cost of obtaining the bSi-based substrates. We developed the ‘recipe’ for fabrication of bSi substrates by supporting SERS signal enhancement at the NIR (785 nm) excitation, which meets biomedical application requirements and does not lead to overheating and thermal damage of the analyte that may occur at the vis-range excitation [35,36,37]. Moreover, uniformity of vertically aligned hot spots on bSi microstructures and the opportunity to control their spatial distribution over large areas open the way towards the detection of extremely low amounts of molecules, though the integration of the signal obtained from the whole substrate area has uniform signal enhancement. We show that bSi based SERS substrates provide a breakthrough in the trade-off between cost and efficiency, making this an important step towards a simple, effective, low-cost and reliable sensing technique.

## 2. Materials and Methods

Single side polished 2“x 275 ± 25 μm p-doped Si <100> wafer with bulk resistivity of 1–30 Ohm∙cm was used for the fabrication of black silicon SERS substrates. To ensure that the surface impurities of the substrate did not affect the fabrication process, the wafers were pre-cleaned with acetone and isopropyl alcohol in an ultrasonic bath at 60 W/20 °C for 10 min and 5 min, respectively. To further ensure that there were no left-over dirt particles, the wafers were oxygen plasma treated for 2 min at 20 sccm flow rate and at a radio frequency (RF) power of 150 W, using Plasma-Etch (PE-50, Plasma Etch Inc.; Carson City, NV, USA).

To obtain bSi with different surface profiles suitable for SERS and to optimize the procedure of bSi fabrication and gold coating, the Bosch etching process was used [38,39]. This cost efficient and relatively fast method allows one to obtain uniformly distributed high aspect ratio relief features. Through the cyclic Bosch process, we can obtain self-masked black silicon structures within a single RIE system under controlled environmental conditions.

The process consists of the following two steps: (i) formation of plasma-polymerized fluorocarbon islands from C_4_F_8_ gas [40] serving as a mask, and (ii) etching using SF_6_/O_2_ gas combination. For the deposition stage, ICP is employed in addition to RIE. This step allows producing high density plasma with low ion bombardment energy at lower pressures needed for micro-masking. Helium backside cooling was used to keep the temperature of the wafer controlled. The deposition/etching parameters used for bSi fabrication are presented in Table 1. The time of etching stage varied in the range of 18–30 min to reveal its influence on the bSi surface geometrical features. Morphological analysis was performed using Zeiss LEO 1550 scanning electron microscope (SEM). All presented images were obtained at 5 kV acceleration voltage using the InLens detector. The SEM images of bSi samples etched at different times are presented in Figure 1. The sizes of related bSi geometrical features are summarized in Table 2. All dimensional characteristics were extracted from SEM images using open-source ImageJ software.

The next step was bSi coating with the gold layer using Q300T Plus magnetron sputter (Quorum Technologies Ltd.; Lewes, UK) at a pressure of 0.001 mbar and sputter current of 80 mA. The bSi samples that were etched at different times and had different surface areas were covered with a layer of gold under the same conditions for which 25 nm of gold on a flat SiO_2_ substrate occurs. Hereinafter, bSi sputtered with Au are labeled as bSi18, bSi25 and bSi30, related to their etching times of 18, 25 and 30 min, respectively.

Further, the bSi/Au substrates were treated with N_2_ to protect them from undesired adsorption on their surface before coating them with the analyte. Quartz slides were sputtered with gold (SiO_2_/Au) under the same conditions as bSi substrates, to be used for comparison. 

Measurements for all fabricated SiO_2_/Au and bSi/Au substrates were performed under the same conditions. The substrates were incubated for 4 h in a solution of 4-mercaptobenzoic acid (4-MBA, HSC_6_H_4_CO_2_H; Sigma; Saint Louis, MO, USA) of a concentration of 0.005 mol/L in ethanol to facilitate the formation of a molecular monolayer on the gold surface of these substrates, through the -SH group. After that, all the samples were washed with ethanol for 10 min so that unbounded 4-MBA molecules were removed. The samples were dried with N_2_ before proceeding with Raman measurements. 

Raman and SERS measurements of 4-MBA (bulk) and 4-MBA self-assembled monolayers (SAM) on SiO_2_/Au and bSi/Au, respectively, were performed by inVia (Renishaw, UK) Raman spectrometer. Two excitation lasers were used, with wavelengths of 785 nm and 514 nm, delivering 5 mW and 3 mW, respectively, on the sample surface at an exposure time of 10 s. The grating 1200 line/mm and a microscope air objective of ×50 magnification, NA 0.75 were utilized. The intensity was recorded as counts per second, and simple baseline correction was performed for better spectral graphical representation. Spectra were collected from at least three points on the sample and then averaged.

To evaluate structural uniformity of the substrates, SERS mapping was performed over an area of 10 × 10 μm^2^ with 1 μm spatial resolution at 785 nm excitation wavelength, delivering 0.3 mW on the sample surface for an exposure time equal to 5 s.

## 3. Experimental Results

Sputtering of the bSi samples with gold resulted in the formation of plasmonic nanostructures of various shapes. Typical silicon-gold structures are presented in Figure 2. Gold nanostructure parameters (thickness, shape, uniformity) are summarized in Table 3. The more developed surface of samples bSi25 and bSi30 contributed to the formation of an almost two-times thinner layer of gold, compared to sample bSi18. The gold layer is non-continuous in all samples; however, some differences between the samples are revealed. In bSi18, the area in the pillar base proximity remains uncovered, but the gold layer is uniform and dense between the rarely distributed thin pillars. In contrast, in case of bSi25 and bSi30, the pillars and their close vicinities are covered by a somewhat-uniform gold pattern, while the interpillar areas are not gold painted (see Figure 2B,D,F). As a result, the Si band is always present in the Raman and SERS spectra, collected from the surfaces. 

This peculiarity of the bSi-based SERS substrates can be employed to determine the concentrations of reagents in solutions. The Raman signal from Si serves as a reference to normalize the SERS signal of the detected molecules at hot spots, in terms of their non-uniform distribution over the surface [41]. The pillars in all three samples are densely covered with gold nanoparticles, which may serve as hot spots for SERS (see Figure 1A,C,E and corresponding inset).

In all three analyzed cases, bSi/Au substrates comprise quasi-cylindrical ‘pillars’ covered with defected gold (enriched with crucks or ‘plasmonic holes’), which might be beneficial for substantial SERS enhancement.

To evaluate the efficiency of the fabricated substrates for SERS applications, we used 4-MBA as a standard test molecule [42]. It exhibits a characteristic Raman spectrum (see Figure 3) and may form a self-assembled monolayer (SAM) of molecules on the surfaces covered with gold, through the formation of covalent bonding S-Au [43]. This process is easily monitored in Raman and SERS spectra as the band 2570 cm^−1^ arising from -SH in 4-MBA disappears in the spectrum of SAM and a new band associated with S-Au at 257 cm^−1^ appears [44]. To reveal the effect of bSi structures in the enhancements of the Raman signal intensity, we used quartz slides sputtered with gold under equivalent conditions, as a reference. The spectrum of SAM of 4-MBA on the SiO_2_/Au obtained at 785 nm excitation is presented in Figure 3. The gold nanostructures formed on the flat SiO_2_ surface gave no Raman intensity enhancement, as no characteristic 4-MBA bands were detected except the Raman spectrum of SiO_2_ substrate itself.

In contrast to the SiO_2_/Au substrate with SAM of 4-MBA, all bSi/Au substrates provided the enhancement of 4-MBA Raman signal. The enhancement of Raman signal occurs when the excitation wavelength coincides with the wavelength of surface plasmons. The bSi/Au substrates with 4-MBA SAMs were analyzed under 514 nm and 785 nm excitation. No resonances were detected for 514 nm excitation, while for the excitation at 785 nm, significant Raman signal enhancement was obtained (see Figure 4A,B for 514 nm and 785 nm excitation, respectively). The characteristic bands in SERS spectra of 4-MBA on bSi/Au substrates and their assignments are summarized in Table 4.

The obtained experimental results are in good agreement with the simulations performed in [45]. The maximum of the absorption cross-section for Si@Au core@shell structures, with characteristic dimensions of 68–88 nm diameter for Si core and 21 nm thickness of the gold shell, is around 780–800 nm, and thinning of the gold layer results in the red shift of the absorption cross-section of nanostructures of this type. These structures correspond to the bSi apex with a gold cap for all bSi samples in our experiments. Moreover, it was demonstrated in [45] that vertical alignment of the groups of gold nanostructures with diameters of 50 nm, which are overlapped with each other and form dumbbell-like structures, have weakly pronounced absorption cross-section maxima of around 500–520 nm and intense absorption cross-section maxima in the NIR range (600–1000 nm). These simulations explain the high 4-MBA SERS signal of molecular monolayers on all the bSi/Au substrates under the 785 nm excitation and support insignificant but observable SERS spectra of 4-MBA molecules detected with 514 nm excitation from bSi18. As it follows from the parameters of the gold layer summarized in Table 3 and simulation results [45], only semi-spheres with a radius of 25 nm exhibit satisfactory absorption at 500–520 nm, which can provide SERS signal enhancement. The decrease in the gold layer thickness, which occurred for the samples bSi25 and bSi30 due to their more developed surface and larger cone-like structures, led to no SERS signal enhancement at 514 nm excitation. However, the vertical alignment of gold nanostructures and silicon core provided bSi25 and bSi30 with efficiency in NIR (see Figure 4).

The substrates were tested for the uniformity of the SERS signal enhancement. All studied bSi substrates are characterized by quite uniform SERS intensity distributions, and the standard deviations for the 1588 cm^−1^ band intensity of 4-MBA molecules were 6.1%, 5.2% and 4.3% (see Figure 5). However, the bSi18 substrate, due to the circumstances described above, provides significantly larger SERS signal enhancement.

## 4. Numerical Simulation

Along with the Si@Au core@shell structures responsible for SERS enhancement in the NIR range, cracks in the thin gold layer covering the silicon pillars (see Figure 2) may give rise to a strong enhancement of the local electric field, due to plasmon hot spots localized on the cracks’ edges. The light absorption of bSi/Au substrates comprising bSi quasi-cylindrical ‘pillars’ covered with a defected (i.e., enriched with plasmonic hot spots) gold layer was studied by averaging the light scattered over the pilar surface. The scattering cross-section of a pillar illuminated by a plane wave along the pillar axis and the polarization in a perpendicular direction to this axis can be described in terms of the dipole moments of hot spots randomly distributed on its surface as the following [47,48]:(1)$$\sigma =\frac{k_{0}{}^{4}}{6\pi \varepsilon_{0}{}^{2}{\left|E_{0}\right|}^{2}}{\left|\sum_{i}p_{i}\right|}^{2}$$
where $p_{i}$ is the dipole moment of the *i*th hot spot, summation is carried out over all hot spots on the pillar, and $E_{0}$ and $k_{0}$ are amplitude and wave number of the incident wave, respectively.

One can observe from Figure 6 that cracks in the gold layer increase the pillar absorption cross-section in the near-IR spectral range, i.e., plasmon hot spots provide up to one order of a magnitude enhancement for the gold-plated pillar at wavelengths longer than 800 nm. However, for all types of bSi geometries, in the visual part of the spectrum, the absorption cross-section of pillars covered with a perfect gold layer slightly exceeds that of the pillars having cracks in the gold coating. By comparing the data, one can see that a decrease in the gold layer thickness from 25 nm to 13 nm can be compensated by the larger size of the pillars (bSi25 and bSi30 geometries in Figure 6).

The electric field distribution along the surface of the gold painted pillars were obtained using Comsol Multiphysics by using parameters (pillar size, distance between pillars, gold thickness and crack size) we observed in the experiment (see Table 3). By considering different distributions of the hot spots over a pillar surface, we calculated the absorption cross-section of an individual pillar. The absorbance of the whole bSi surface was obtained by using periodic boundary conditions. The values and dispersion law of silicon refraction index and extinction coefficient were taken from [49].

Figure 7 shows absorbance of an array of gold covered silicon pillars with different distributions of cracks. One can see that in the wavelength range 800–2000 nm, the covering of bSi with a defected gold layer increases the absorbance because of local field enhancement at the gold layer defects. Specifically, in the NIR spectral range, the absorbance of bSi covered with cracked gold is 4–8 times higher than that of samples comprising an array of smooth bSi/Au pillars.

For all samples, in the vicinity of the reflection suppression ‘threshold’ wavelength (500–750 nm), the measured reflectivity was 30–35% higher than those obtained by numerical simulation. Such a discrepancy originates from the uncontrollable doping of silicon in the course of RIE [50] that was not taken into account in the numerical simulation, aiming at a qualitative rather than quantitative description of the reflectance spectra.

## 5. Discussion

Both numerical and experimental results demonstrate that sputtering with very a thin (approx. 10 nm) but cracked gold layer allows one to achieve stronger—in comparison with ‘pyramidal’ bSi [45]—enhancement of the Raman signal. Importantly, silicon outperforms cheaper glass materials as a plasmonic nanostructures support because the etch rate of glass is very low and is only 1/40th of the Si etch rate with fluorine radicals [51]. This makes micro-structuring of glass by ICR-RIE much more complex and time-consuming than bSi fabrication. Glasses are inhomogeneous, both chemically and structurally, and this inhomogeneity significantly complicates 3D or even 2D patterning, while the charging effects, which accompany the etching process of dielectrics, may drastically influence the reaction progress.

In terms of the choice of fabrication method, RT ICP-RIE also outperforms other techniques, including MACE, which requires using gold (Au), silver (Ag), aluminum (Al), copper (Cu) and nickel (Ni) catalysts [52]. In the one-step MACE process, metal on the surface emerges from the etching aqueous solutions, containing metal nitrates, while in the two-step MACE process, metals are physically deposited on the silicon surface before the chemical etching step [53]. After the etching step, a metal catalyst moves inside the Si wafer and locates itself at the bottom of silicon structures (deep in the pore), preventing formation of the 3D distribution of hot spots. However, such a 3D arrangement of the plasmonic hot spots on the bSi surface increases (in comparison with 2D distribution) the probability for the molecules of the analyte to meet a hot spot, thus significantly improving the sensitivity. Moreover, we demonstrate that cracks in the gold layer and gaps between the deposited gold islands provide significant enhancement of the Raman signal.

To sum up, we demonstrated that bSi offers the cheapest, most efficient, and reliable SERS substrate. The cost reduction was achieved by (i) the use of RT ICP-RIE instead of more complex and expensive cryogenic ICP-RIE, and (ii) the reduction in gold layer thickness. Until very recently, all the proposed SERS substrate fabrication techniques were based on very thick noble metal layers. For instance, in [54], the Ag layer thickness was 150 nm to achieve satisfactory Raman signal. In [55], the authors used a 100 nm thick layer of gold (RT ICP-RIE), in [56], the gold layer thickness was even larger and amounted to 400 nm (cryo-ICP-RIE), and in [57], the thickness of the deposited aluminum layer was around 1000 nm (ICP-RIE, temperature not stated). However, when the noble metal layer is of hundreds of nanometers, the 3D packaging of plasmonic nanostructures does not occur. Black silicon only serves as a supporting material for 2D nanoparticle distribution, so that the substrate could be replaced by another, easy to manufacture rough material. In the bSi structures fabricated and analyzed in this study, the geometry of the bSi is used to provide efficient Raman signal enhancement in full.

## 6. Conclusions

We fabricated the black Si substrate with the controllable surface profile covered in a ten nanometers thick gold layer. The SERS measurements with 4-MBA revealed that the fabricated substrates allow one to obtain satisfactory Raman signal enhancement with excellent spatial uniformity (less than 6% signal intensity variation over a 20 × 20 μm^2^ area) at the NIR excitation. The obtained results indicate that bSi can be employed for scalable fabrication of affordable SERS substrates, which may become a stable, simple and reliable consumable material in laboratory routine, providing relatively high sensitivity at low cost.

Numerical simulation was performed to calculate the absorbance of bSi/Au samples by considering the geometrical features of silicon pillars and non-perfect coverage with gold. According to modeling results, for all geometries of bSi surface relief, we observed a substantial increase in the NIR absorption cross-section of individual bSi pillars covered with the defected gold, in which cracks work as plasmonic hot spots. For more developed bSi surfaces (e.g., bSi25 and bSi30 in Table 3), even the sputtering of the structure with very thin, <15 nm, but cracked gold allows us to achieve strong Raman signal due to significant enhancement of the local field in the vicinity of the densely distributed hot spots. 

## Figures and Tables

**Figure 1 materials-16-01948-f001:**
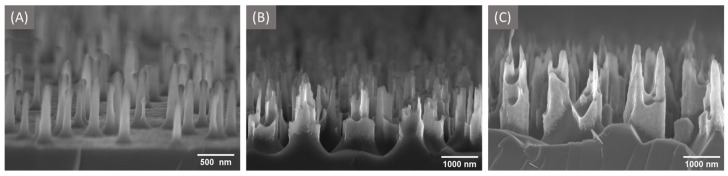
The micro-structured silicon surface obtained at the etching time of (**A**) 18 min, (**B**) 25 min, (**C**) 30 min.

**Figure 2 materials-16-01948-f002:**
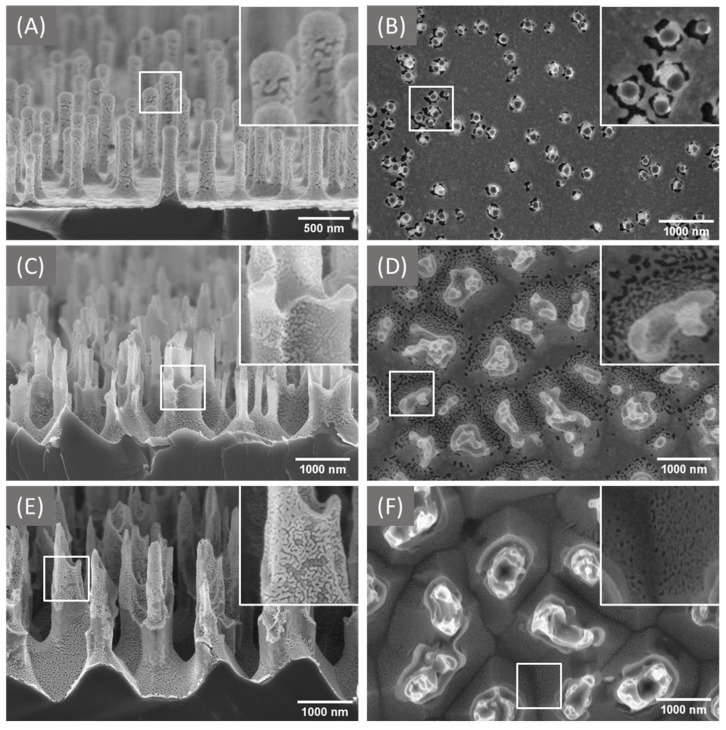
SEM side- and top-view images of the bSi18 (**A**,**B**), bSi25 (**C**,**D**), bSi30 (**E**,**F**), sputtered with Au. Inset images (**B**–**F**) are 800 × 800 nm and (**A**) is 400 × 400 nm.

**Figure 3 materials-16-01948-f003:**
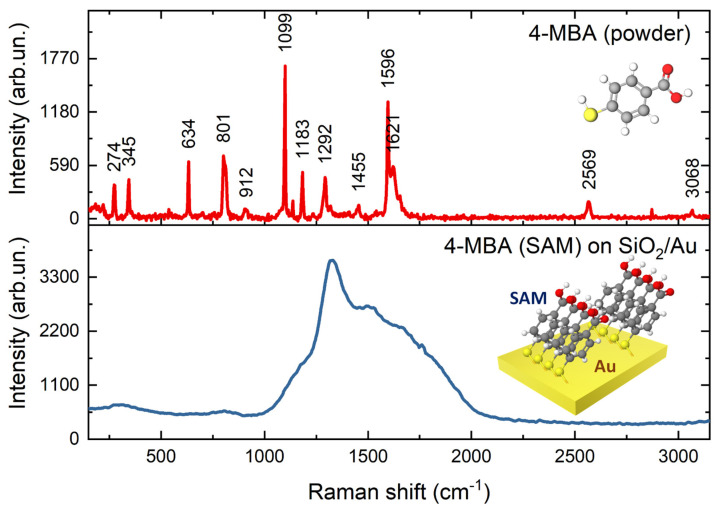
Raman spectra of 4-MBA powder and SAM of 4-MBA molecules on SiO_2_/Au. Measurements were performed at the following conditions: λ_ex_ = 785 nm, laser power = 3 mW, accumulation time = 10 s. 4-MBA composition: Gray—carbon atoms, red—oxygen atoms, yellow—sulfur atom, white—hydrogen atoms.

**Figure 4 materials-16-01948-f004:**
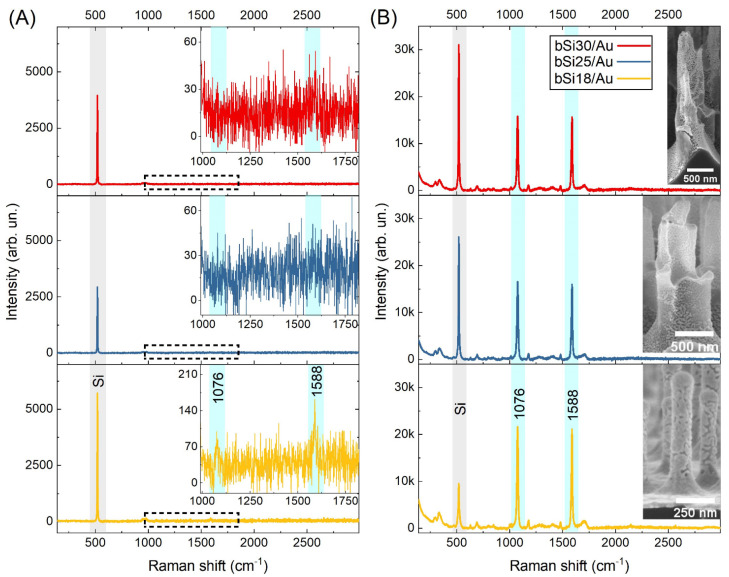
SERS spectra of 4-MBA on (**A**) 514 nm (10 s, 5 mW) and (**B**) 785 nm (10 s, 3 mW) excitation wavelengths on bSi/Au samples.

**Figure 5 materials-16-01948-f005:**
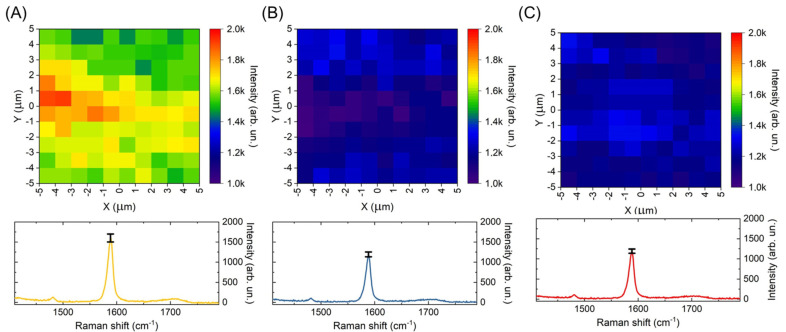
Spatial distributions of SERS signal on 1588 cm^−1^ for 4-MBA on (**A**) bSi18/Au, (**B**) bSi25/Au and (**C**) bSi30/Au samples.

**Figure 6 materials-16-01948-f006:**
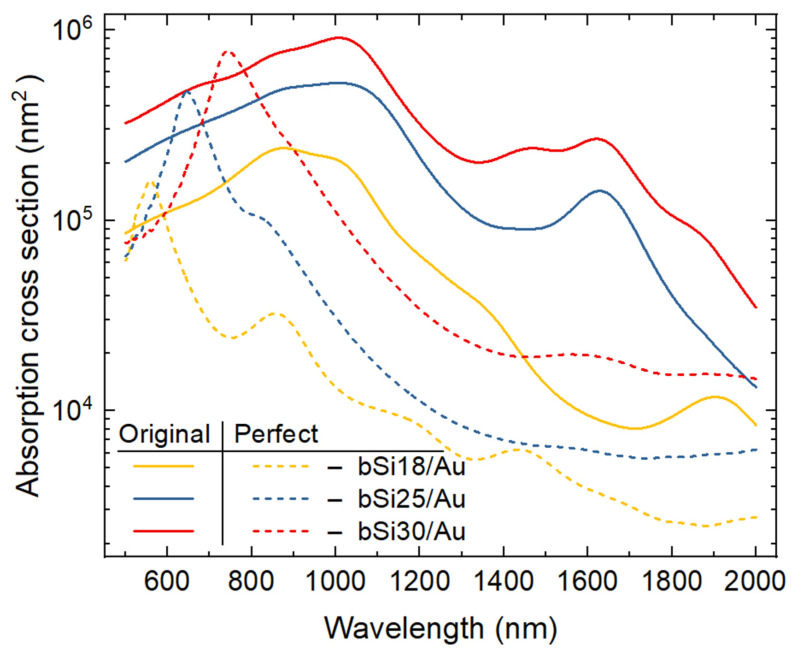
Absorption cross-section of individual pillars of bSi18, bSi25 and bSi30 geometry (taken from Table 3), either coated with perfect gold layer without cracks (perfect, dashed lines) or by defected gold-comprising cracks (original, solid lines).

**Figure 7 materials-16-01948-f007:**
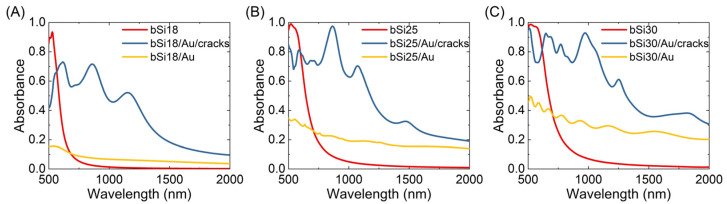
Calculated absorbance spectra for array of pillars of bare, covered with continuous and cracked gold layer bSi of the following three different geometries: bSi18 (**A**), bSi25 (**B**) and bSi30 (**C**). Geometrical parameters are taken from Table 3. Silicon refractive index, n, and extinction coefficient, k, were taken in the intervals 4–3.46 and 0.03–4.2 × 10^−8^, respectively, for the wavelengths 500–2000 nm [49] in the simulations.

**Table 1 materials-16-01948-t001:** Deposition/etching parameters for bSi fabrication.

	Gasses	Flow (sccm)	Time (min)	RF (W)	ICP (W)	Pressure (mTorr)	DC Bias (V)	Temperature (°C)	He-Backside Cooling (sccm)
Deposition	C_4_F_8_/SF_6_	40:1	1	20	295	20	100	20	5
Etching	SF_6_/O_2_	10:9	18–30	30	0	30	100	20	10

**Table 2 materials-16-01948-t002:** Geometrical parameters of structures covering the Si substrate at different etching times.

Etching Time (min)	Height (nm)	Apex Thickness (nm)	Base Thickness (nm)	Pillars Density, pcs/μm^2^
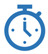	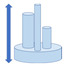	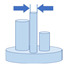	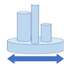	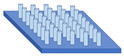
18	639 ± 126	112 ± 21	110 ± 32	4.33
25	1625 ± 298	125 ± 36	638 ± 155	1.12
30	2217 ± 433	144 ± 63	832 ± 377	0.56

**Table 3 materials-16-01948-t003:** Parameters of the structures and surfaces of the bSi.

Sample	Thickness of Au Layer (nm)	Type of Geometry	Area Uncovered with Au (%)
bSi18	25 ± 5	Cylindrical pillars	8.24 (98 μm^2^ was analyzed)
bSi25	14 ± 5	Cylindrical pillars on stump	9.03 (49 μm^2^ was analyzed)
bSi30	13 ± 4	Spiked mountains	Difficult to analyze. Very mosaic structure

**Table 4 materials-16-01948-t004:** Assignment of the characteristic bands of both Raman spectra of bulk 4-MBA and SERS spectra of SAM of 4-MBA on bSi18/Au [45,46].

4-MBA Bulk (cm^−1^)	4-MBA on bSi18/Au (cm^−1^)	Peak Assignments
	257 (w)	ν(Au-S)
–	520 (m)	Si (E_2g_ mode)
634 (m) #	632 (w)	ν_6b_ + δ(COOH) deformation
–	691 (m)	γ(CH) out of plane
801 (m)	–	ν_10a_ in-plane aromatic ring breathing mode + -COH deformation
–	848 (w)	β(COO^–^)
912 (w)	–	β(SH)
1099 (s)	1076 (s)	ν_12_(_a1_) in-plane aromatic ring breathing mode + ν(CS)
1183 (m)	1182 (w)	ν_9_(_a1_) + δ(CH) deformation
1292 (m)	–	ν_3_
1455 (w)	1481 (w)	ν(CC) + γ(CH)
1596 (s)	1588 (s)	ν_8a_(_a1_) totally symmetric aromatic ring vibration
1621 (m)	–	ν(C=O), ν(OC) + δ(HOC)
–	1706 (m)	ν(C=O), COOH are hydrogen bonded [46]
2569 (m)	–	ν(SH) stretching mode

# The vibrational modes are as follows: ν—stretching; β—bending; δ—deformation; γ—out of plane deformation. (w), (m) and (s) indicate ‘weak’, ‘medium’ and ‘strong’ bands in the spectra, respectively.

## Data Availability

Data are available from the correspondent author by reasonable request.

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
