# Peer review of "Black Silicon: Breaking through the Everlasting Cost vs. Effectivity Trade-Off for SERS Substrates"

_materials, 2023, doi:10.3390/ma16051948_

Round 1
Reviewer 1 Report
The manuscript is technically sound and meet the scope of the journal. However, the paper lacks enough novelty as SERS using RIE bSi has been demonsrated many times before:
https://koreascience.kr/article/JAKO202123563820597.page
https://onlinelibrary.wiley.com/doi/full/10.1002/andp.201300035
https://www.sciencedirect.com/science/article/pii/S0956566313006507
https://aip.scitation.org/doi/full/10.1063/5.0084907
https://scholar.google.com/scholar?as_sdt=0%2C5&q=Surface+enhanced+Raman+scattering+RIE+black+silicon&btnG=
...
1. Please further elaborate the novelty of this paper in the context of similar papers above (and probably more).
2. Discussion on the cost issue. Another approach is to use optimized MACE bSi substrate since it does not require any tool and/or use other metal films such as Ag or Al.
Author Response
Dear Reviewer,
We are thankful for your comments and valuable suggestions that have allowed us to improve presentation of the obtained results. Following Reviewer’s suggestions, we change the title of the revised manuscript to “Black silicon: breaking through the everlasting cost vs. effectivity trade-off for SERS substrates”.
Our point-to-point responses to the Reviewers comments are given below. The relevant changes in a revised manuscript are highlighted in yellow. Typos and misprints were corrected throughout the texts and are also marked in yellow.
Sincerely yours,
On behalf of all authors
Prof. Polina Kuzhir,
Department of Physics and Mathematics,
Center for Photonics Sciences,
University of Eastern Finland, Joensuu, Finland
Reviewer #1
Comments and Suggestions for Authors:
The manuscript is technically sound and meet the scope of the journal. However, the paper lacks enough novelty as SERS using RIE bSi has been demonstrated many times before:
https://koreascience.kr/article/JAKO202123563820597.page
https://onlinelibrary.wiley.com/doi/full/10.1002/andp.201300035
https://www.sciencedirect.com/science/article/pii/S0956566313006507
https://aip.scitation.org/doi/full/10.1063/5.0084907
https://scholar.google.com/scholar?as_sdt=0%2C5&q=Surface+enhanced+Raman+scattering+RIE+black+silicon&btnG=
- Please further elaborate the novelty of this paper in the context of similar papers above (and probably more).
Our Response:
The authors are thankful to the Reviewer for this remark. In our paper we demonstrate that the fabricated bSi substrate outperform other in terms of cost/efficiency trade-off. The cost of the bSi based substrate can be reduced by either using of RT ICP-RIE instead of cryogenic ICP-RIE or by the reduction of the gold layer thickness. For now, very thick noble metal layers are conventionally used. For instance, in [1] the Ag layer thickness was 150 nm to achieve satisfactory Raman signal. In [2] the authors used a 100 nm thick layer of gold (RT ICP-RIE), in [3] the gold layer thickness was even larger and amounted to 400 nm (cryo-ICP-RIE), and in [4] the thickness of the deposited aluminum layer was around 1000 nm (ICP-RIE temperature not stated). When the noble metal layer is of hundreds of nanometers, the role of the height on the bSi structures vanishes and 3D packaging of plasmonic nanostructures does not occur. In such a case bSi only serves as a supporting material and can be replaced by virtually any substrate provided it has a sufficient roughness.
We added relevant discussion to the text of the manuscript on Pages 10-11, Lines 328-341, and refer the papers suggested by the Reviewer.
“To sum up, we demonstrated that the bSi offers the cheapest, very efficient, and reliable SERS substrate. The cost reduction was achieved by (i) the use of RT ICP-RIE instead of more complex and expensive cryogenic ICP-RIE, and (ii) the reduction of the gold layer thickness. Until very recently, all the proposed SERS substrate fabrication techniques are based on very thick noble metal layers. For instance, in [54] the Ag layer thickness was 150 nm to achieve satisfactory Raman signal. In [55] the authors used a 100 nm thick layer of gold (RT ICP-RIE), in [56] the gold layer thickness was even larger and amounted to 400 nm (cryo-ICP-RIE), and in [57] the thickness of the deposited aluminum layer was around 1000 nm (ICP-RIE, temperature not stated). But when the noble metal layer is of hundreds of nanometers, the 3D packaging of plasmonic nanostructures does not occur. Black silicon only serves as a supporting material for 2D nanoparticle distribution, so that the substrate could be replaced by another easi-er-to-manufacture rough material. In the bSi structures, fabricated and analyzed in this study, the geometry of the bSi is used to provide efficient Raman signal enhancement in full.”
References:
- Kim, H.J.; Kim, B.; Lee, D.; Lee, B.-H.; Cho, C. Fabrication of Surface-Enhanced Raman Scattering Substrate Using Black Silicon Layer Manufactured through Reactive Ion Etching. J. Sens. Sci. Technol. 2021, 30, 267–272, doi:10.46670/jsst.2021.30.4.267.
- Gervinskas, G.; Seniutinas, G.; Hartley, J.S.; Kandasamy, S.; Stoddart, P.R.; Fahim, N.F.; Juodkazis, S. Surface-Enhanced Raman Scattering Sensing on Black Silicon. Ann. Phys. 2013, 525, 907–914, doi:10.1002/andp.201300035.
- 56. Deng, Y.L.; Juang, Y.J. Black Silicon SERS Substrate: Effect of Surface Morphology on SERS Detection and Application of Single Algal Cell Analysis. Bioelectron. 2014, 53, 37–42, doi:10.1016/j.bios.2013.09.032.
- Lin, B.W.; Tai, Y.H.; Lee, Y.C.; Xing, D.; Lin, H.C.; Yamahara, H.; Ho, Y.L.; Tabata, H.; Daiguji, H.; Delaunay, J.J. Aluminum-Black Silicon Plasmonic Nano-Eggs Structure for Deep-UV Surface-Enhanced Resonance Raman Spectroscopy. Appl. Phys. Lett. 2022, 120, 51102, doi:10.1063/5.0084907.
- Discussion on the cost issue. Another approach is to use optimized MACE bSi substrate since it does not require any tool and/or use other metal films such as Ag or Al.
Our response:
The authors are thankful to the Reviewer for this interesting suggestion. However, we would like to admit that the Metal-Assisted Chemical Etching (MACE) method of bSi production cannot allow one to avoid the step of Au or Ag deposition on the already etched bSi surface. MACE approach requests using gold (Au), silver (Ag), aluminum (Al), copper (Cu) and nickel (Ni) on the surface of bSi as a catalyst [1]. In the one-step MACE process, metal on the surface emerges from the etching aqueous solutions, containing metal nitrates, while in the two-step MACE, metals are physically deposited on the silicon surface before the chemical etching step [2]. After the etching step, metal catalyst moves inside the Si wafer and locates at the bottom of silicon structures (deep in the pore). But such metal nanoparticle distribution does not allow one to obtain 3D distribution of noble metal “hot spots”, which spread from the bottom to the apex of each structure. Volume distribution of the “hot spots” (in contrast to 2D distribution) increases the probability for the molecules of the analyte to meet the “hot spot”, thus significantly improving the sensitivity. Moreover, as we demonstrate in our paper, cracks in the gold layer and gaps between the deposited gold islands provide significant enhancement of the Raman signal, which is possible only in case of high density of neighboring nanoparticles. And finally, only Au and Ag exhibit effective plasmonic resonance in vis-to-NIR range and, thus, provide good signal enhancement. Considering all the above, it becomes clear that there is no cost reduction in the case of MACE in comparison with RT-ICP-RIE.
Corresponding discussion is added to Page 10, Lines 316-327.
- Alhmoud, H.; Brodoceanu, D.; Elnathan, R.; Kraus, T.; Voelcker, N.H. A MACEing Silicon: Towards Single-Step Etching of Defined Porous Nanostructures for Biomedicine. Prog. Mater. Sci. 2021, 116, 100636, doi:10.1016/J.PMATSCI.2019.100636.
- Arafat, M.Y.; Islam, M.A.; Mahmood, A.W. Bin; Abdullah, F.; Nur-E-Alam, M.; Kiong, T.S.; Amin, N. Fabrication of Black Silicon via Metal-Assisted Chemical Etching—a Review. Sustain. 2021, 13, doi:10.3390/su131910766.
d Conclusion sections to Pages 8-11.

Reviewer 2 Report
This manuscript investigates the use of black silicon and localized surface plasmon structures in a hybrid system for surface enhanced Raman spectroscopy (SERS). The system they have designed presents enhanced SERS signals at near-infrared (785 nm) excitation, fulfilling the requirements for biomedical applications and avoiding overheating and thermal damage. In my view, this work is very similar to what the authors previously published in Ref. [36] and lacks sufficient novelty, or at least failed to address them. Therefore, the authors should clearly discuss about the difference between this work and their published one at the beginning of the paper and provide a detailed explanation of the new findings presented in this work. Additionally, there are several other areas in the manuscript that could be improved, as listed below.
(1) Title is ambiguous. It could be improved by adding more context or rephrasing to provide a better understanding of what is meant by "everlasting trade-off" and the role of "black silicon" as a solution.
(2) I suggest avoiding the abbreviation "bSi" for black silicon as it can be confused with the chemical formula for another material. Instead, use "black silicon" or "b-Si."
(3) The authors state in the manuscript that black silicon can be used for absorbing UV light, however, they fail to include any references to back up their claim.
(4) The authors should provide more detailed references and comparisons to previous works on the implementation of SERS using black silicon. Specifically, when discussing various approaches to SERS implementation based on metallic nanoparticles or thin films, the authors should cite relevant previous works.
(5) The English is understandable, but there are minor deficiencies which should be corrected eventually. Additionally, several sentences require modification to improve clarity. For instance, the meaning of this sentence is not clear to the reader: “However, strong dependence of SPR on the geometry of plasmonic structures [21], [22], [23] and their three- dimensional organization implies bSi profile control. From the other hand, this disadvantage may be diminished by low production cost and if gold consumption will be reduced.”
Author Response
Dear Reviewer,
We are thankful for your comments and valuable suggestions that have allowed us to improve presentation of the obtained results. Following Reviewer’s suggestions, we change the title of the revised manuscript to “Black silicon: breaking through the everlasting cost vs. effectivity trade-off for SERS substrates”.
Our point-to-point responses to the Reviewers comments are given below. The relevant changes in a revised manuscript are highlighted in yellow. Typos and misprints were corrected throughout the texts and are also marked in yellow.
Sincerely yours,
On behalf of all authors
Prof. Polina Kuzhir,
Department of Physics and Mathematics,
Center for Photonics Sciences,
University of Eastern Finland, Joensuu, Finland
Reviewer #2
Comments and Suggestions for Authors:
This manuscript investigates the use of black silicon and localized surface plasmon structures in a hybrid system for surface enhanced Raman spectroscopy (SERS). The system they have designed presents enhanced SERS signals at near-infrared (785 nm) excitation, fulfilling the requirements for biomedical applications and avoiding overheating and thermal damage. In my view, this work is very similar to what the authors previously published in Ref. [36] and lacks sufficient novelty, or at least failed to address them. Therefore, the authors should clearly discuss about the difference between this work and their published one at the beginning of the paper and provide a detailed explanation of the new findings presented in this work.
Our Response:
We are thankful to the Reviewer for paying attention to this important point. You are right, we had to clearly explain the difference in the task statements of the published ACS AMI 2020 paper [10.1021/acsami.0c13570] paper and current manuscript.
The ACS AMI 2020 paper [10.1021/acsami.0c13570] was the first one in the foreseen series of manuscripts devoted to systematic analysis of various factors responsible for the efficiency of the bSi based SERS substrates. Among them are micro- and nano-geometry of Si surface, Si doping level, and the features of the deposited gold ‘layer’ like its homogeneity, integrity, existence of core-shell Si/gold particles of different shapes, sizes and orientations, gold micro and nano-cracks, etc.
It is known that any change in the bSi fabrication process leads to changes of its surface geometry. Depending on the application, one may target microstructures which absorb effectively in NIR, or visible or UV light. It is determined by shape and size, and density of the Si ‘outgrowths’, their roughness, and nano-structural features. The latter also determines the peculiarities of gold deposition, leading to either continuous film formation, or to defected or even densely distributed individual nanoparticles Si surface coverage.
The main goal of the ACS AMI 2020 paper [10.1021/acsami.0c13570] was to apply bSi of specific geometry with pyramidal relief having hydrophilic properties which is of special importance to the sensing of living cells. This type of bSi could not be fabricated in a one-step room temperature etching process. The analysis of electromagnetic performance was done with the focus on geometrical features of silicon/Au particles inherent in specific fabrication conditions supporting hydrophilicity. In that case hot spots were core-shell gold Si-spheres and dumbbells with a dominating vertical orientation along bSi/ Au cones. Finite elements modelling and SEM image analysis demonstrated that this orientation is crucial for achieving the high efficiency of the substrate in the NIR spectral range.
The present manuscript solves another task. We intended to find an optimal bSi geometry, which could be achieved through the one-step room temperature RIE process and at the same time is efficient for SERS. We reached that through combination of this specific geometry with exceptionally thin layer of gold, but we did not address the hydrophilicity of the bSi substrate. The final goal is determining conditions suitable for scalable fabrication of affordable SERS substrates, which may become a stable, simple in use and reliable consumable material in laboratory routine providing relatively high sensitivity for various analytes at low-cost.
In this paper, we found theoretically and proved experimentally that when sputtered with very thin (approx. 10 nm) yet cracked gold layer, such substrates are capable to provide compatible - in comparison with ‘pyramidal’ bSi [10.1021/acsami.0c13570] sputtered with thicker gold (20-50 nm) - enhancement of Raman signal.
Our original finding, i.e., gold layer defects as a tool to approach better SERS performance even with extremely thin gold layer, provides alternative insight on the way of achieving good Raman signal enhancement and differs from the conclusions of most published papers [50-53, mentioned in Manuscript], in which it is stated that the thicker the gold layer is, the higher SERS enhancement is achieved. Along with a feasible and simple bSi fabrication recipe it supports our manuscript novelty.
The corresponding discussion is added on Page 10, Lines 306-308.
“Both numerical and experimental results demonstrate that sputtering with very thin (approx. 10 nm) but cracked gold layer allows one to achieve stronger – in comparison with ‘pyramidal’ bSi [45] – enhancement of the Raman signal.”
And Pages 10-11, Lines 328-341:
“To sum up, we demonstrated that the bSi offers the cheapest, very efficient, and reliable SERS substrate. The cost reduction was achieved by (i) the use of RT ICP-RIE instead of more complex and expensive cryogenic ICP-RIE, and (ii) the reduction of the gold layer thickness. Until very recently, all the proposed SERS substrate fabrication techniques are based on very thick noble metal layers. For instance, in [54] the Ag layer thickness was 150 nm to achieve satisfactory Raman signal. In [55] the authors used a 100 nm thick layer of gold (RT ICP-RIE), in [56] the gold layer thickness was even larger and amounted to 400 nm (cryo-ICP-RIE), and in [57] the thickness of the deposited aluminum layer was around 1000 nm (ICP-RIE, temperature not stated). But when the noble metal layer is of hundreds of nanometers, the 3D packaging of plasmonic nanostructures does not occur. Black silicon only serves as a supporting material for 2D nanoparticle distribution, so that the substrate could be replaced by another easier-to-manufacture rough material. In the bSi structures, fabricated and analyzed in this study, the geometry of the bSi is used to provide efficient Raman signal enhancement in full.”
References:
- Kim, H.J.; Kim, B.; Lee, D.; Lee, B.-H.; Cho, C. Fabrication of Surface-Enhanced Raman Scattering Substrate Using Black Silicon Layer Manufactured through Reactive Ion Etching. J. Sens. Sci. Technol. 2021, 30, 267–272, doi:10.46670/jsst.2021.30.4.267.
- 55. Gervinskas, G.; Seniutinas, G.; Hartley, J.S.; Kandasamy, S.; Stoddart, P.R.; Fahim, N.F.; Juodkazis, S. Surface-Enhanced Raman Scattering Sensing on Black Silicon. Phys. 2013, 525, 907–914, doi:10.1002/andp.201300035.
- 56. Deng, Y.L.; Juang, Y.J. Black Silicon SERS Substrate: Effect of Surface Morphology on SERS Detection and Application of Single Algal Cell Analysis. Bioelectron. 2014, 53, 37–42, doi:10.1016/j.bios.2013.09.032.
- 57. Lin, B.W.; Tai, Y.H.; Lee, Y.C.; Xing, D.; Lin, H.C.; Yamahara, H.; Ho, Y.L.; Tabata, H.; Daiguji, H.; Delaunay, J.J. Aluminum-Black Silicon Plasmonic Nano-Eggs Structure for Deep-UV Surface-Enhanced Resonance Raman Spectroscopy. Phys. Lett. 2022, 120, 51102, doi:10.1063/5.0084907.
Additionally, there are several other areas in the manuscript that could be improved, as listed below.
(1) Title is ambiguous. It could be improved by adding more context or rephrasing to provide a better understanding of what is meant by "everlasting trade-off" and the role of "black silicon" as a solution.
Our Response:
The authors agree with the Reviewer. We rephrased the title to remove the ambiguity. The rephrased title is set as: “Black silicon: breaking through the everlasting cost vs. effectivity trade-off for SERS substrates”.
(2) I suggest avoiding the abbreviation "bSi" for black silicon as it can be confused with the chemical formula for another material. Instead, use "black silicon" or "b-Si."
Our Response:
Thank you for this advice. Unfortunately, there is no commonly accepted abbreviation for black silicon. In the literature one can find BSi ([1], [2]), b-Si ([3], [4]), BS ([5]), and bSi ([6]-[9]). We used “bSi” to be consistent with our previous papers. We believe that this should not confuse a reader.
- Arafat, M.Y.; Islam, M.A.; Mahmood, A.W. Bin; Abdullah, F.; Nur-E-Alam, M.; Kiong, T.S.; Amin, N. Fabrication of Black Silicon via Metal-Assisted Chemical Etching—a Review. Sustain. 2021, 13, doi:10.3390/su131910766.
- Liu, X.; Coxon, P.R.; Peters, M.; Hoex, B.; Cole, J.M.; Fray, D.J. Black Silicon: Fabrication Methods, Properties and Solar Energy Applications. Energy Environ. Sci. 2014, 7, 3223–3263, doi:10.1039/C4EE01152J.
- Chai, J.Y.H.; Wong, B.T.; Juodkazis, S. Black-Silicon-Assisted Photovoltaic Cells for Better Conversion Efficiencies: A Review on Recent Research and Development Efforts. Mater. Today Energy 2020, 18, 100539, doi:10.1016/j.mtener.2020.100539.
- Hu, F.; Dai, X.-Y.; Zhou, Z.-Q.; Kong, X.-Y.; Sun, S.-L.; Zhang, R.-J.; Wang, S.-Y.; Lu, M.; Sun, J. Black Silicon Schottky Photodetector in Sub-Bandgap near-Infrared Regime. Opt. Express 2019, 27, 3161, doi:10.1364/oe.27.003161.
- Atteia, F.; Le Rouzo, J.; Berginc, G.; Simon, J.-J.; Escoubas, L. Black Silicon (BS) Using Room-Temperature Reactive Ion Etching (RT-RIE) for Interdigitated Back Contact (IBC) Silicon Solar Cells. In Proceedings of the https://doi.org/10.1117/12.2509326; SPIE, February 27 2019; Vol. 10913, p. 29.
- Ali, M.; Zhou, F.; Chen, K.; Kotzur, C.; Xiao, C.; Bourgeois, L.; Zhang, X.; MacFarlane, D.R. Nanostructured Photoelectrochemical Solar Cell for Nitrogen Reduction Using Plasmon-Enhanced Black Silicon. Nat. Commun. 2016, 7, 1–5, doi:10.1038/ncomms11335.
- Yuan, H.C.; Yost, V.E.; Page, M.R.; Stradins, P.; Meier, D.L.; Branz, H.M. Efficient Black Silicon Solar Cell with a Density-Graded Nanoporous Surface: Optical Properties, Performance Limitations, and Design Rules. Appl. Phys. Lett. 2009, 95, 123501, doi:10.1063/1.3231438.
- Fan, Z.; Cui, D.; Zhang, Z.; Zhao, Z.; Chen, H.; Fan, Y.; Li, P.; Zhang, Z.; Xue, C.; Yan, S. Recent Progress of Black Silicon: From Fabrications to Applications. Nanomaterials 2021, 11, 1–26, doi:10.3390/nano11010041.
- Golubewa, L.; Karpicz, R.; Matulaitiene, I.; Selskis, A.; Rutkauskas, D.; Pushkarchuk, A.; Khlopina, T.; Michels, D.; Lyakhov, D.; Kulahava, T.; et al. Surface-Enhanced Raman Spectroscopy of Organic Molecules and Living Cells with Gold-Plated Black Silicon. ACS Appl. Mater. Interfaces 2020, 12, 50971–50984, doi:10.1021/acsami.0c13570.
(3) The authors state in the manuscript that black silicon can be used for absorbing UV light, however, they fail to include any references to back up their claim.
Our Response:
The authors are grateful to the reviewer for careful reading of the manuscript. The proper citation has been missed accidentally. UV-vis to NIR light absorbance from 250 nm to 2500 nm was demonstrated in, e.g., [1] and [2]. Corresponding references are added to Page 1, Lines 25-26.
- Lv, J.; Zhang, T.; Zhang, P.; Zhao, Y.; Li, S. Review Application of Nanostructured Black Silicon. Nanoscale Res. Lett. 2018, 13, 1–10, doi:10.1186/s11671-018-2523-4.
- Yuan, H.C.; Yost, V.E.; Page, M.R.; Stradins, P.; Meier, D.L.; Branz, H.M. Efficient Black Silicon Solar Cell with a Density-Graded Nanoporous Surface: Optical Properties, Performance Limitations, and Design Rules. Appl. Phys. Lett. 2009, 95, 123501, doi:10.1063/1.3231438.
(4) The authors should provide more detailed references and comparisons to previous works on the implementation of SERS using black silicon. Specifically, when discussing various approaches to SERS implementation based on metallic nanoparticles or thin films, the authors should cite relevant previous works.
Our Response:
The authors are thankful to the Reviewer for this remark. We added corresponding discussion to the text of the manuscript on Pages 10-11, Lines 328-341:
“To sum up, we demonstrated that the bSi offers the cheapest, very efficient, and reliable SERS substrate. The cost reduction was achieved by (i) the use of RT ICP-RIE instead of more complex and expensive cryogenic ICP-RIE, and (ii) the reduction of the gold layer thickness. Until very recently, all the proposed SERS substrate fabrication techniques are based on very thick noble metal layers. For instance, in [54] the Ag layer thickness was 150 nm to achieve satisfactory Raman signal. In [55] the authors used a 100 nm thick layer of gold (RT ICP-RIE), in [56] the gold layer thickness was even larger and amounted to 400 nm (cryo-ICP-RIE), and in [57] the thickness of the deposited aluminum layer was around 1000 nm (ICP-RIE, temperature not stated). But when the noble metal layer is of hundreds of nanometers, the 3D packaging of plasmonic nanostructures does not occur. Black silicon only serves as a supporting material for 2D nanoparticle distribution, so that the substrate could be replaced by another easier-to-manufacture rough material. In the bSi structures, fabricated and analyzed in this study, the geometry of the bSi is used to provide efficient Raman signal enhancement in full.”
References:
- Kim, H.J.; Kim, B.; Lee, D.; Lee, B.-H.; Cho, C. Fabrication of Surface-Enhanced Raman Scattering Substrate Using Black Silicon Layer Manufactured through Reactive Ion Etching. J. Sens. Sci. Technol. 2021, 30, 267–272, doi:10.46670/jsst.2021.30.4.267.
- 55. Gervinskas, G.; Seniutinas, G.; Hartley, J.S.; Kandasamy, S.; Stoddart, P.R.; Fahim, N.F.; Juodkazis, S. Surface-Enhanced Raman Scattering Sensing on Black Silicon. Phys. 2013, 525, 907–914, doi:10.1002/andp.201300035.
- 56. Deng, Y.L.; Juang, Y.J. Black Silicon SERS Substrate: Effect of Surface Morphology on SERS Detection and Application of Single Algal Cell Analysis. Bioelectron. 2014, 53, 37–42, doi:10.1016/j.bios.2013.09.032.
- 57. Lin, B.W.; Tai, Y.H.; Lee, Y.C.; Xing, D.; Lin, H.C.; Yamahara, H.; Ho, Y.L.; Tabata, H.; Daiguji, H.; Delaunay, J.J. Aluminum-Black Silicon Plasmonic Nano-Eggs Structure for Deep-UV Surface-Enhanced Resonance Raman Spectroscopy. Phys. Lett. 2022, 120, 51102, doi:10.1063/5.0084907.
(5) The English is understandable, but there are minor deficiencies which should be corrected eventually. Additionally, several sentences require modification to improve clarity. For instance, the meaning of this sentence is not clear to the reader: “However, strong dependence of SPR on the geometry of plasmonic structures [21], [22], [23] and their three- dimensional organization implies bSi profile control. From the other hand, this disadvantage may be diminished by low production cost and if gold consumption will be reduced.”
Our Response:
The authors are grateful to the reviewer for deep examination of the paper. Several corrections are made throughout the text and marked in yellow. According to the sentences, mentioned by the Reviewer, an unfortunate mistake was made when formatting the text of the article. The second sentence is redundant. It has been removed in the revised version.
Corresponding corrections are made on Page 2, Line 80.

Reviewer 3 Report
The work is devoted to the formation of 3D plasmonic structures based on black silicon for SERS applications. The authors propose to use the method of ion-plasma etching of black silicon to form nanospires for their subsequent decoration with gold nanoparticles. This approach makes it possible to increase the concentration of hot spots due to the larger effective area and uneven formation of nanoparticles during magnetron sputtering. This approach can undoubtedly be considered as a fast and relatively uncomplicated process in the creation of SERS sensors. The work was done at a good level, all methods and approaches are well described. From the point of view of the SERS method, the work could be more interesting if the authors demonstrated the limits of detection of analytes.
Author Response
Dear Reviewer,
We are thankful for your comments and valuable suggestions that have allowed us to improve presentation of the obtained results. Following Reviewer’s suggestions, we change the title of the revised manuscript to “Black silicon: breaking through the everlasting cost vs. effectivity trade-off for SERS substrates”.
Our point-to-point responses to the Reviewers comments are given below. The relevant changes in a revised manuscript are highlighted in yellow. Typos and misprints were corrected throughout the texts and are also marked in yellow.
Sincerely yours,
On behalf of all authors
Prof. Polina Kuzhir,
Department of Physics and Mathematics,
Center for Photonics Sciences,
University of Eastern Finland, Joensuu, Finland
Reviewer #3
Comments and Suggestions for Authors:
The work is devoted to the formation of 3D plasmonic structures based on black silicon for SERS applications. The authors propose to use the method of ion-plasma etching of black silicon to form nanospires for their subsequent decoration with gold nanoparticles. This approach makes it possible to increase the concentration of hot spots due to the larger effective area and uneven formation of nanoparticles during magnetron sputtering. This approach can undoubtedly be considered as a fast and relatively uncomplicated process in the creation of SERS sensors. The work was done at a good level, all methods and approaches are well described. From the point of view of the SERS method, the work could be more interesting if the authors demonstrated the limits of detection of analytes.
Our Response:
The authors are grateful for the high evaluation of the manuscript. However, the detection limit is good to demonstrate using some other molecules than 4-mba (e.g., those not forming the covalent bonding with the substrate) and for this reason was not in the scope of the present study. Nevertheless, further bSi structure optimization according to our current findings will be employed to solve applied problems and limits of detection will be necessarily provided.

Reviewer 4 Report
The article develops a new technique for producing microstructured silicon, which has an important application value. A sufficiently large number of studies of the geometry and various optical properties of the obtained layers were carried out, a large number of microphotographs are given. Raman spectra of samples were studied. Even a small numerical simulation of the absorption properties of microstructured silicon (bSi) has been performed, which is qualitatively consistent with the experimental results obtained. Overall, the paper makes a good impression and can be published.
Author Response
Thanks a lot for the good attitude of our paper.
Reviewer 5 Report
The manuscript describes the use of silicon substrates whose surface has been nanostructured, as a template for the deposition of gold films and their subsequent use as SERS substrates. Their performance is illustrated be further coating them with the SERS standard MBA. There are many issues related to the study and presentation which need to be considered before it can be considered for publication;
(i) As a general issue, there are a number of companies who have commercialised SERS substrates, made I believe by similar methods (e.g. https://www.silmeco.com/) . The authors should discuss these, and indicate how their method improves on them.
(ii) In relation to (i), the authors should also comment on why for example glass cannot be similarly etched to provide a template for gold deposition.
(iii) Abstract: "Hybrid statistical/numerical simulation proves the substantial ..." - the simulation may "support" but it does not "prove"
(iv) Abstract: "detection of trace amounts of analytes," What is meant by "trace amounts"? The authors should be quantitative.
(v) The Introduction should very briefly indicate the origin of the changes in the optical properties of the processed silicon surface. Specular reflectance is obviously reduced, and diffuse reflectance/scattering increased. Is the electronic structure changed because of surface/defect states? Does this reduce the band gap? Introduce mid bandgap defect states?
(vi) References are missing in several places:
"..the surface plasmon resonance (SPR) results in an increase of the Raman scattering cross-section by several orders of magnitude allowing one to detect trace amounts of molecules adsorbed on the interface."
"The former usually give significant enhancement of Raman intensity, however, are non-stable" - reference required,
"storage condition-dependent," - reference required
"inconvenient in routine use," - reference required
"requiring immobilization on the flat surface for measuring"- reference required
"poorly reproducible and nonuniform." - reference required
"Namely, SERS substrate fabrication is always a trade-off between stability, cost, efficiency, reliability, uniformity, density of hot-spots and scalability." - this statement requires considerable elaboration, with substantiating references, as it is a major aspect of the title of the article, and is not justified elsewhere.
(vii) "This, in its turn, will cause the increase of the Raman signal of the adsorbed molecules of the analyte." - why - is this simply due to the surface structure, or is there any electronic interaction? Would a similarly structures glass substrate do the same?
(viii) "...(i) the polymer deposition" - the authors should explain what polymer this is.
(ix) The Methods section should explain how the electron microscopy was performed, how the dimensions of the features were determined, how the thickness and uniformity of the gold coatings were determined.
(x) UV/vis/NIR Absorption/Reflection spectra of each bSi substrate should be shown, before and after gold deposition.
(xi) "To reveal the effect of bSi structures in the enhancements of the Raman signal intensity, we used quartz slides sputtered with gold under equivalent conditions as a reference." - for any meaningful comparison, the quartz slides should be subjected to the same processing as the Si substrates.
(xii) "No resonances were detected for 514 nm excitation,..." see (x) Absorption/Reflection spectra must be shown
(xiii) "Raman spectra of bulk 4-MBA" - what is bulk 4-MBA and how was it measured? This should be described in the methods.
(xiv) "The obtained experimental results are in a good agreement with the simulations, performed in [36]." - The simulations initially compute the absorption spectrum - this should be compared this a spectrum of the fabricated structures. "These simulations explain high 4-MBA SERS signal of molecular monolayers..." - The authors should compare the absorption/reflection spectra with the simulated spectra.
(xv) "The cracks in the gold film, which covers silicon pillars (see Figure 2), can result in significant enhancement of the local field..." This is the first mention of this, and if it is an important aspect of the study it should be mentioned in the Abstract, Introduction and Methodology. There is explicit experimental demonstration to support the statment.
Author Response
Dear Reviewer,
We are thankful for your comments and valuable suggestions that have allowed us to improve presentation of the obtained results. Following Reviewer’s suggestions, we change the title of the revised manuscript to “Black silicon: breaking through the everlasting cost vs. effectivity trade-off for SERS substrates”.
Our point-to-point responses to the Reviewers comments are given below. The relevant changes in a revised manuscript are highlighted in yellow. Typos and misprints were corrected throughout the texts and are also marked in yellow.
Sincerely yours,
On behalf of all authors
Prof. Polina Kuzhir,
Department of Physics and Mathematics,
Center for Photonics Sciences,
University of Eastern Finland, Joensuu, Finland
Reviewer #5
Comments and Suggestions for Authors:
The manuscript describes the use of silicon substrates whose surface has been nanostructured, as a template for the deposition of gold films and their subsequent use as SERS substrates. Their performance is illustrated be further coating them with the SERS standard MBA. There are many issues related to the study and presentation which need to be considered before it can be considered for publication;
(i) As a general issue, there are a number of companies who have commercialised SERS substrates, made I believe by similar methods (e.g., https://www.silmeco.com/). The authors should discuss these, and indicate how their method improves on them.
Our Response:
The authors are thankful to the Reviewer for this comment. We added a discussion about silicon structures obtained by similar fabrication procedures and pointed the main disadvantages of our approach. We drastically reduced the amount of gold needed and demonstrated that in combination with highly developed surface of black silicon even 10-nm thick gold layer with cracks provides significant Raman signal enhancement. Moreover, room temperature ICP-RIE we employed significantly simplifies the fabrication procedure and reduces its cost. From a comparison of the substrates developed by us with the commercial platforms mentioned by the Reviewer, it follows that our substrates have several additional characteristics that are superior to the commercial version (https://www.silmeco.com/). Commercially available silicon nanopillar substrates are virtually disposable because pillars are damaged and stuck together after contact with the analyte, i.e., they cannot be reused in the laboratory routine and require careful handling. Our substrates are stable, simple and low-cost and re-usable being affordable for wide use and high-quality research.
The following discussion is added to Pages 10-11, Lines 328-341:
“To sum up, we demonstrated that the bSi offers the cheapest, very efficient, and reliable SERS substrate. The cost reduction was achieved by (i) the use of RT ICP-RIE instead of more complex and expensive cryogenic ICP-RIE, and (ii) the reduction of the gold layer thickness. Until very recently, all the proposed SERS substrate fabrication techniques are based on very thick noble metal layers. For instance, in [54] the Ag layer thickness was 150 nm to achieve satisfactory Raman signal. In [55] the authors used a 100 nm thick layer of gold (RT ICP-RIE), in [56] the gold layer thickness was even larger and amounted to 400 nm (cryo-ICP-RIE), and in [57] the thickness of the deposited aluminum layer was around 1000 nm (ICP-RIE, temperature not stated). But when the noble metal layer is of hundreds of nanometers, the 3D packaging of plasmonic nanostructures does not occur. Black silicon only serves as a supporting material for 2D nanoparticle distribution, so that the substrate could be replaced by another easier-to-manufacture rough material. In the bSi structures, fabricated and analyzed in this study, the geometry of the bSi is used to provide efficient Raman signal enhancement in full.”
References:
- Kim, H.J.; Kim, B.; Lee, D.; Lee, B.-H.; Cho, C. Fabrication of Surface-Enhanced Raman Scattering Substrate Using Black Silicon Layer Manufactured through Reactive Ion Etching. J. Sens. Sci. Technol. 2021, 30, 267–272, doi:10.46670/jsst.2021.30.4.267.
- 55. Gervinskas, G.; Seniutinas, G.; Hartley, J.S.; Kandasamy, S.; Stoddart, P.R.; Fahim, N.F.; Juodkazis, S. Surface-Enhanced Raman Scattering Sensing on Black Silicon. Phys. 2013, 525, 907–914, doi:10.1002/andp.201300035.
- 56. Deng, Y.L.; Juang, Y.J. Black Silicon SERS Substrate: Effect of Surface Morphology on SERS Detection and Application of Single Algal Cell Analysis. Bioelectron. 2014, 53, 37–42, doi:10.1016/j.bios.2013.09.032.
- 57. Lin, B.W.; Tai, Y.H.; Lee, Y.C.; Xing, D.; Lin, H.C.; Yamahara, H.; Ho, Y.L.; Tabata, H.; Daiguji, H.; Delaunay, J.J. Aluminum-Black Silicon Plasmonic Nano-Eggs Structure for Deep-UV Surface-Enhanced Resonance Raman Spectroscopy. Phys. Lett. 2022, 120, 51102, doi:10.1063/5.0084907.
(ii) In relation to (i), the authors should also comment on why for example glass cannot be similarly etched to provide a template for gold deposition.
Our Response:
Micro-structuring of glass by ICR-RIE method is much more complex and long procedure. For instance, the etch rate of glass is very low and is only 1/40th of Si etch rate when etching process occurs with fluorine radicals [1]. Glasses are inhomogeneous both chemically and structurally, and this inhomogeneity complicates 3D or even 2D pattering of the surface significantly. Moreover, since glass is a dielectric, the charging effects, which accompany the etching process, may drastically influence the reaction progress. Optical properties of micro-structured glass materials differ significantly, are characterized by high transmission rate, and are more applicable for diffractive optical element devices [2].
Corresponding discussion is added to Page 10, Lines 308-315.
Reference:
- 1. Weigel, C.; Brokmann, U.; Hofmann, M.; Behrens, A.; Rädlein, E.; Hoffmann, M.; Strehle, S.; Sinzinger, S. Perspectives of Reactive Ion Etching of Silicate Glasses for Optical Microsystems. Opt. Microsystems 2021, 1, 1–22, doi:10.1117/1.jom.1.4.040901.
- Tang, Y.H.; Lin, Y.H.; Chen, P.L.; Shiao, M.H.; Hsiao, C.N. Comparison of Optimised Conditions for Inductively Coupled Plasma-Reactive Ion Etching of Quartz Substrates and Its Optical Applications. Micro Nano Lett. 2014, 9, 395–398, doi:10.1049/mnl.2014.0093.
(iii) Abstract: "Hybrid statistical/numerical simulation proves the substantial ..." - the simulation may "support" but it does not "prove"
Our Response:
The text is corrected according to the Reviewer’s recommendation.
Corresponding changes are added on Page 1, Line 18.
(iv) Abstract: "detection of trace amounts of analytes," What is meant by "trace amounts"? The authors should be quantitative.
Our Response:
The authors are thankful to the Reviewer for this comment. In the present study no concentration limit detection was performed as it was not in the scope of the study. However, to exclude the possible misunderstanding of the essence of the results achieved we rephrased the last sentence of the Abstract.
Corresponding corrections are added to Page 1, Abstract.
(v) The Introduction should very briefly indicate the origin of the changes in the optical properties of the processed silicon surface. Specular reflectance is obviously reduced, and diffuse reflectance/scattering increased. Is the electronic structure changed because of surface/defect states? Does this reduce the band gap? Introduce mid bandgap defect states?
Our Response:
The origin of the decrease of specular reflectance and the increase of absorbance/scattering is beyond the scope of the present study.
Relevant discussion is added to Page 1, Lines 33-40 in the revised manuscript:
“The suppression of the specular reflectance and enhancement of absorbance can be described in terms of the structural defects in the Si lattice or impurities (specifically sulfur) which appear during the surface micro-structuring [7]. These defects and impurities create IR absorbing states in the band gap and are also responsible for the trapping of the photoexcited carriers (for flat silicon band gap is 1.07 eV) [9]. The absorbance caused by these defects is enhanced by multiple reflections between the microstructures of bSi, which, in turn, suppresses specular reflection and increases absorbance.”
References:
- Wu, C.; Crouch, C.H.; Zhao, L.; Carey, J.E.; Younkin, R.; Levinson, J.A.; Mazur, E.; Farrell, R.M.; Gothoskar, P.; Karger, A. Near-Unity below-Band-Gap Absorption by Microstructured Silicon. Appl. Phys. Lett. 2001, 78, 1850–1852, doi:10.1063/1.1358846.
- Zhong, H.; Ilyas, N.; Song, Y.; Li, W.; Jiang, Y. Enhanced Near-Infrared Absorber: Two-Step Fabricated Structured Black Silicon and Its Device Application. Nanoscale Res. Lett. 2018, 13, 1–8, doi:10.1186/s11671-018-2741-9.
(vi) References are missing in several places:
Our Response:
(a) "..the surface plasmon resonance (SPR) results in an increase of the Raman scattering cross-section by several orders of magnitude allowing one to detect trace amounts of molecules adsorbed on the interface." - reference required
References: [17, 18], Page 2, Line 54
- Pérez-Jiménez, A.I.; Lyu, D.; Lu, Z.; Liu, G.; Ren, B. Surface-Enhanced Raman Spectroscopy: Benefits, Trade-Offs and Future Developments. Chem. Sci. 2020, 11, 4563–4577, doi:10.1039/d0sc00809e.
- Wang, J.; Lin, W.; Cao, E.; Xu, X.; Liang, W.; Zhang, X. Surface Plasmon Resonance Sensors on Raman and Fluorescence Spectroscopy. Sensors (Switzerland) 2017, 17, doi:10.3390/s17122719.
(b) "The former usually give significant enhancement of Raman intensity, however, are non-stable" - reference required,
References: [25] Page 2, Line 71
- Mo, S.; Shao, X.; Chen, Y.; Cheng, Z. Increasing Entropy for Colloidal Stabilization. Sci. Rep. 2016, 6, 1–7, doi:10.1038/srep36836.
(c) "storage condition-dependent," - reference required
References: [26] Page 2, Line 71
- Shi, L.; Zhang, L.; Tian, Y. Rational Design of Surface‐Enhanced Raman Scattering Substrate for Highly Reproducible Analysis. Anal. Sens. 2022, e202200064, doi:10.1002/anse.202200064
(d) "inconvenient in routine use," - reference required
References: [27] Page 2, Line 72
- 27. Khnykina, K.A.; Baranov, M.A.; Babaev, A.A.; Dubavik, A.Y.; Fedorov, A. V; Baranov, A. V; Bogdanov, K. V Comparison Study of Surface-Enhanced Raman Spectroscopy Substrates. Phys. Conf. Ser. 2021, 1984, 12020, doi:10.1088/1742-6596/1984/1/012020.
(e) "requiring immobilization on the flat surface for measuring"- reference required
References: [28] Page 2, Line 72
- Mikac, L.; Gotić, M.; Gebavi, H.; Ivanda, M. The Variety of Substrates for Surface-Enhanced Raman Spectroscopy. 2017 IEEE 7th Int. Conf. Nanomater. Appl. Prop. N. 2017, 2017-Janua, doi:10.1109/NAP.2017.8190187.
(f) "poorly reproducible and nonuniform." - reference required
References: [27, 28] Page 2, Line 73
- Khnykina, K.A.; Baranov, M.A.; Babaev, A.A.; Dubavik, A.Y.; Fedorov, A. V; Baranov, A. V; Bogdanov, K. V Comparison Study of Surface-Enhanced Raman Spectroscopy Substrates. J. Phys. Conf. Ser. 2021, 1984, 12020, doi:10.1088/1742-6596/1984/1/012020.
- 28. Mikac, L.; Gotić, M.; Gebavi, H.; Ivanda, M. The Variety of Substrates for Surface-Enhanced Raman Spectroscopy. 2017 IEEE 7th Int. Conf. Nanomater. Appl. Prop. N. 2017, 2017-Janua, doi:10.1109/NAP.2017.8190187.
"Namely, SERS substrate fabrication is always a trade-off between stability, cost, efficiency, reliability, uniformity, density of hot-spots and scalability." - this statement requires considerable elaboration, with substantiating references, as it is a major aspect of the title of the article, and is not justified elsewhere.
Our Response:
The authors are thankful to the Reviewer for this valuable comment. We added corresponding references and discussion in the text to Pages 2, 9, 10.
(vii) "This, in its turn, will cause the increase of the Raman signal of the adsorbed molecules of the analyte." - why - is this simply due to the surface structure, or is there any electronic interaction? Would a similarly structures glass substrate do the same?
Our Response:
The Reviewer raised a very interesting problem. The enhancement of the Raman signal occurs through the interaction of the molecules of the analyte with gold nanostructures through the electromagnetic mechanism mediated by the surface plasmon resonance and enhanced by numerous reflections in between the silicon microstructures. As it is already mentioned in our answer to the question # (v) multiple sulfur impurities and defect states, which occur in the Si during the micro-structuring, also contribute to the Raman signal enhancement via absorption increase.
Glass structures do not produce such effects and can serve only as a 3D matrix for gold nanoparticle distribution. However, as it was already mentioned in our answer to the Reviewer’s question #(i), fused silica (SiO2) is a much more difficult material for ICP-RIE micro-structuring, e.g., it is characterized by very low etch rate which is only 1/40th of Si when etching with fluorine radicals [1].
Reference:
- Weigel, C.; Brokmann, U.; Hofmann, M.; Behrens, A.; Rädlein, E.; Hoffmann, M.; Strehle, S.; Sinzinger, S. Perspectives of Reactive Ion Etching of Silicate Glasses for Optical Microsystems. J. Opt. Microsystems 2021, 1, 1–22, doi:10.1117/1.jom.1.4.040901.
(viii) "...(i) the polymer deposition" - the authors should explain what polymer this is.
Our Response:
Thank you for paying our attention to this issue. The process includes two steps: (i) formation of plasma-polymerized fluorocarbon islands from C4F8 gas [40], which serves as a mask, and (ii) etching using SF6/O2 gas combination. We revise the manuscript text (Page 3, Lines 117-118) accordingly:
“The process consists of two steps: (i) formation of plasma-polymerized fluorocarbon islands from C4F8 gas [40] serving as a mask, and (ii) etching using SF6/O2 gas com-bination.”
Reference:
- Labelle, C.B.; Opila, R.; Kornblit, A. Plasma Deposition of Fluorocarbon Thin Films from C-C4F8 Using Pulsed and Continuous Rf Excitation. J. Vac. Sci. Technol. A Vacuum, Surfaces, Film. 2005, 23, 190–196, doi:10.1116/1.1830496.
(ix) The Methods section should explain how the electron microscopy was performed, how the dimensions of the features were determined, how the thickness and uniformity of the gold coatings were determined.
Our Response:
The authors are thankful to the Reviewer for careful reading the manuscript. Corresponding information is added to Materials and Methods section to Page 3, Lines 124-129.
“Morphological analysis was performed using Zeiss LEO 1550 scanning electron microscope (SEM). All presented images were obtained at 5 kV acceleration voltage using InLens detector.” … “The sizes of related bSi geometrical features are summarized in Table 2. All dimensional characteristics were extracted from SEM images using open-source ImageJ software.”
(x) UV/vis/NIR Absorption/Reflection spectra of each bSi substrate should be shown, before and after gold deposition.
Our Response:
Thank you for this advice. The bSi reflection suppression ‘threshold’ wavelength depends on the shape and, especially, on the height of the silicon relief. In our case the threshold wavelength for bSi18, bSi25 and bSi30 is around 520, 580 and 630 nm (see Figure 1) in attached pdf file.
Figure 1. Calculated absorbance spectra for array of pillars of bare, covered with continuous and cracked gold layer bSi of 3 different geometries: bSi18 (A), bSi25 (B) and bSi30 (C). Geometrical parameters are taken from Table 3. Refractive index n and extinction coefficient k were taken in the interval 4 – 3.46 and 0.03 – 4.2 10-8 for the wavelengths 500-2000nm [1] in the simulations.
All the measured by Perkin Elmer Lambda 18 spectrophotometer reflection coefficients are 30-35% higher than that obtained in the numerical simulations in the frequency range around reflection suppression ‘threshold’ wavelength (500-750nm) due to the discrepancy of silicon doping level in the simulation and in real samples. This is because significant additional doping of silicon surface occurred because of RIE [2]) was not considered in the simulations, which were aiming at qualitative rather than quantitative description of the reflectance spectra.
We added corresponding discussion and more detailed explanation of theoretical estimations and modeling results to the Manuscript text (Page 8-11, Lines 253-304).
Reference:
- Schinke, C.; Christian Peest, P.; Schmidt, J.; Brendel, R.; Bothe, K.; Vogt, M.R.; Kröger, I.; Winter, S.; Schirmacher, A.; Lim, S.; et al. Uncertainty Analysis for the Coefficient of Band-to-Band Absorption of Crystalline Silicon. AIP Adv. 2015, 5, 067168, doi:10.1063/1.4923379.
- Lin, Y.J.; Chu, Y.L. Effect of Reactive Ion Etching-Induced Defects on the Surface Band Bending of Heavily Mg-Doped p-Type GaN. Appl. Phys. 2005, 97, 104904, doi:10.1063/1.1894580.
(xi) "To reveal the effect of bSi structures in the enhancements of the Raman signal intensity, we used quartz slides sputtered with gold under equivalent conditions as a reference." - for any meaningful comparison, the quartz slides should be subjected to the same processing as the Si substrates.
Our Response:
Quartz slides sputtered with gold were processed in the same conditions as bSi substrates, i.e., they were incubated in 4-MBA water solutions for the same time, dried, and measurements were performed under 785 nm excitation.
To avoid misunderstanding in the revised manuscript, corresponding information is added in the Methods and Materials section to Page 4, Lines 145-149.
(xii) "No resonances were detected for 514 nm excitation,..." see (x) Absorption/Reflection spectra must be shown
Our Response:
The authors are thankful for the comment. However, the absence of resonance when excited with 514 nm laser follows from the absence of the Raman signal enhancement and does not require any additional spectra demonstration.
(xiii) "Raman spectra of bulk 4-MBA" - what is bulk 4-MBA and how was it measured? This should be described in the methods.
Our Response:
In our paper, ‘bulk” means “powder”. Raman spectra measured from the “bulk” 4-MBA are compared with the SERS spectra of 4-MBA molecules adsorbed on the bSi substrate. However, all measurements are performed in the same conditions as those applied for measuring other substrates to provide reliable comparison and evaluation of enhancement factor, for instance. We added some additional explanation to the Methods and Materials section to Page 4, Lines 155-156.
(xiv) "The obtained experimental results are in a good agreement with the simulations, performed in [36]." - The simulations initially compute the absorption spectrum - this should be compared this a spectrum of the fabricated structures. "These simulations explain high 4-MBA SERS signal of molecular monolayers..." - The authors should compare the absorption/reflection spectra with the simulated spectra.
Our Response:
This issue has been addressed in our answer to query (x) above.
(xv) "The cracks in the gold film, which covers silicon pillars (see Figure 2), can result in significant enhancement of the local field..." This is the first mention of this, and if it is an important aspect of the study it should be mentioned in the Abstract, Introduction and Methodology. There is explicit experimental demonstration to support the statment.
Our Response:
Corresponding discussions and explanations are added to Abstract (Page 1), Introduction (Page 2), Discussion and Conclusion sections to Pages 8-11.

Round 2
Reviewer 5 Report
The manuscript has been significantly improved on the basis of the issues raised in the original review